# Evaluation of Toxicity on *Ctenopharyngodon idella* Due to Tannery Effluent Remediated by Constructed Wetland Technology

**Sobia Ashraf [1], Muhammad Naveed [1,*], Muhammad Afzal [2], Sana Ashraf [3,4], Sajid Rashid Ahmad [3], Khadeeja Rehman [2], Zahir Ahmad Zahir [1] and Avelino Núñez-Delgado [5]**

[1] Institute of Soil and Environmental Sciences (ISES), University of Agriculture, Faisalabad 38000, Pakistan; sobiaashraf13@gmail.com (S.A.); zazahir@yahoo.com (Z.A.Z.)

[2] Soil and Environmental Biotechnology Division (SEBD), National Institute for Biotechnology and Genetic Engineering (NIBGE), Faisalabad 38000, Pakistan; manibge@yahoo.com (M.A.); khadeeja.abc31@gmail.com (K.R.)

[3] College of Earth and Environmental Sciences (CEES), University of the Punjab, Lahore 53700, Pakistan; sana.cees@pu.edu.pk (S.A.); sajidpu@yahoo.com (S.R.A.)

[4] Department of Environmental Sciences, Lahore College for Women University (LCWU), Lahore 54000, Pakistan

[5] Department Soil Science and Agricultural Chemistry, Engineering Polytechnic School, Campus University Lugo, University of Santiago de Compostela, 27002 Lugo, Spain; avelino.nunez@usc.es

* Correspondence: muhammad.naveed@uaf.edu.pk

**Abstract:** Aquatic pollution caused by industrial effluents is an environmental issue, imposing deleterious impacts on the overall environment, specifically, on humans, by disrupting the balance of the ecosystem. Among all the industries, tanneries are considered some of the most polluting due to heavy use of toxic organic and inorganic compounds during leather processing, most of which find their way into rivers, lakes, and streams, thus exerting adverse effects on aquatic life, particularly on fish. Considering the huge concentrations of pollutants present in tannery effluents, toxicity evaluation is of prime importance. Therefore, bioassays are usually employed to assess the acute toxicity of industrial effluents and efficiency of effluent clean-up technologies as they provide a thorough response of test species to the substances present in the tested media. In the present study, the toxic effects of tannery effluent on common grass carp (*Ctenopharyngodon idella*) were studied for 96 h in laboratory conditions. The effluent was added at different concentrations, before and after treatment by constructed wetlands (CWs). During this period, mortality data was collected to calculate the 96 h-LC50 (lethal concentration inducing 50% mortality) and acute toxicity of *C. idella*. In addition to this, observations on change in morphological, physiological, and behavioural patterns were also made every 24 h. The present toxicity assay revealed that the raw tannery effluent changed the morphology, physiology, and behavioural response of fish. Moreover, fish exposure to raw/untreated effluent caused high acute toxicity and 100% mortality, due to the presence of high concentrations of salts and chromium (Cr) metal. While treatment of tannery effluent by CWs vegetated with different plants (*B. mutica*, *L. fusca*, and *T. domingensis*) significantly reduced its toxicity and fish mortality as well, and inoculation of salt and Cr-tolerant endophytic bacteria (*Enterobacter* sp. HU38, *Microbacterium arborescens* HU33, and *Pantoea stewartii* ASI11) further reduced (up to 90%) its toxicity level. Hence, the use of CWs for tannery effluent treatment can be recommended to favour public health and promote the overall safety of the environment.

**Keywords:** biotoxicity test; constructed wetlands; treated tannery effluent; morphological changes; behavioural response; *Ctenopharyngodon idella*

## 1. Introduction

Industrial growth has negatively impacted the environmental quality [1], particularly aquatic life, due to disposal of voluminous amounts of effluents into natural water resources, on a daily basis, all over the world [2–4]. Some chemicals contained in industrial effluents have been reported to be highly toxic, in a variable degree depending on the dose and exposure duration [5,6], having potential to impart serious damage to aquatic life [7,8]. Among the most vulnerable sectors is aquaculture, which is also of main importance, as it constitutes an essential opportunity to meet the global food security challenge [9].

Among all industries, tanneries are considered some of the most polluting, and are a major contributor to aquatic contamination [10]. Tannery effluent is extremely hazardous due to high organic loadings, solids, and metals [11], particularly chromium (Cr), a well-known ubiquitous pollutant considered as a great menace to aquatic environments, and ultimately, human beings [12]. The dark colouration and low oxygen content of tannery effluents indicates the strength of pollution, which ultimately affect the survival of aquatic organisms, especially fish, thus having great ecological relevance [5].

According to Srivastava et al. [13], 1–10% of tannery effluents can kill fish. Almost all types of fish are quite sensitive to polluted environment, and pollutants found in tannery effluent cause significant damage to their physiological and biochemical processes, such as endocrine disruption [14,15]. Huge physiological, histological, haematological, behavioural, genetic, biochemical, and immunological alterations in aquatic organisms have been reported due to Cr exposure [12]. The biological changes in fish caused by pollutants are called biomarkers and can be used for environmental risk assessment [16,17].

Remediation of toxic industrial effluents is crucial for the protection of receiving water bodies [18]. The quality of these obnoxious effluents is most commonly monitored by analysing major pollution parameters such as pH (Pondus Hydrogenii), electrical conductivity (EC), dissolved oxygen (DO), biochemical oxygen demand ($BOD_5$), chemical oxygen demand (COD), total organic carbon (TOC), total dissolved solids (TDS), total suspended solids (TSS), and metals [19–25]. However, these parameters cannot be used to evaluate toxic effects on receiving waters and subsequent susceptibility of aquatic organisms to these toxic pollutants.

The best way to evaluate effluent toxicity effect is to use biotoxicity tests that provide the complete response of the test organisms to all the pollutants in the surrounding environment [19,26–30]. Different organisms such as fish, algae, bacteria, invertebrates, fungi, etc., are most commonly used in the biotoxicity tests to determine the acceptable and safe level of pollutants in industrial discharge [19,30,31].

Generally, fish have been considered as a useful index for the purity of water, hence they can be used in toxicity assays to detect aquatic hazards [32]. Currently, more attention is being given to acute toxicity testing to detect harmful effects of toxic pollutants present in industrial effluents on aquatic organisms, due to the imposition of stringent environmental laws and regulations on industrial discharge [33,34].

In this context, several studies have reported the harmful effects of industrial effluents on fish, such as that by Martin-Skilton et al. [35], who studied some alterations in hepatic biotransformation enzymes as well as reduced levels of testosterone in juvenile turbot (*Scolphthalmus maximus*) on exposure to fuel oil, which elicits threats to the reproductive system of exposed individuals. In another study by Saborido-Ray et al. [36], tested individuals exhibited reduced growth and feed consumption when exposed to higher concentrations of commercial petroleum fuel. The petroleum fuel also negatively affected the growth and survival of the freshwater fish *Oreochromis niloticus* [37].

Besides the petroleum industry, effluent of textile factories also has damaging effects on fish, such as high lipid peroxide levels in fish, which may cause physiological problems when consumed by humans, as reported by Mahabub-uz-zaman et al. [38]. So, conducting investigations regarding quality of fish sold in local markets are crucial, especially in third world countries. Fish toxicity assays by Bhattacharya et al. [39] revealed a 100% survival rate after 72 h exposure of *Poecilia* sp. in ceramic membrane-treated effluent, as compared to 80% survival rate in untreated effluent. Moreover, high

pollutants concentrations can cause rapid suffocation in fish by destruction of the gill epithelium and inhibition of metabolic activities [40].

Morphological and physiological variations, such as loss of body balance, higher rate of mucus secretion, decrease in breathing rate, haemoglobin percentage, red blood cells (RBC) count, and breakage of deoxyribonucleic acid (DNA) were observed in fish exposed to Cr containing industrial effluents by Bakshi and Panigrahi [12]. Industrial effluents also caused significant histopathological deterioration in gill, kidney, liver, and intestine of test organisms along with significant changes in total glycogen, protein, lipid content in gill, muscle, and liver tissues. In addition to this, physiological disorders, such as hypertension, sporadic fever, renal damage, cramps, and malfunctioning in fish general health conditions due to direct discharge of tannery effluent into water channels, have also been reported by Karthikeyan et al. [41].

Constructed wetlands (CWs) are a contrasted technology, useful for remediation of highly polluted industrial effluents, and can be easily managed with low operation and maintenance cost. These biogeophysical engineered systems consisting of mainly vegetation, soils, and their associated microbial assemblages are constructed in such a way to mimic all the physical, chemical, and biological processes occurring in natural wetlands, but within more controlled environmental conditions that sustain strong plant–substrate–microbe interactions. Many studies have reported its performance for degradation/detoxification of organic and inorganic contaminants present in tannery effluent [42–44], but none of the studies have evaluated the resulting reduction in toxicity level of effluent. The present work is an endeavour to study the effect of tannery effluent (untreated and treated) on the survival of locally available fish *Ctenopharyngodon idella,* commonly known as "grass carp", which is widely found in natural water bodies in Pakistan. In view of that, this study has a clear novelty, and would allow to obtain new data of environmental relevance.

## 2. Materials and Methods

### 2.1. Collection of Fish Samples

*Ctenopharyngodon idella*, irrespective of sex, were collected from local fish farms at Satiana Road, Faisalabad (Pakistan), in closed polythene bags filled with oxygen and transported to the laboratory. These fish were kept in a large glass aquarium, filled with fresh water, at $26 \pm 1\ °C$, in the fish toxicology laboratory, Soil and Environmental Biotechnology Division, National Institute for Biotechnology and Genetic Engineering (NIBGE), Faisalabad. An aeration pump was used to ensure the continuous supply of oxygen in the aquarium, essential for fish survival. Fish were acclimatized under these conditions for one month, prior to their exposure to tannery effluent. During this period, the fish were fed with commercial fish pellets, and water was renewed on alternate days to remove the remains of feed and faecal matter/metabolic waste.

### 2.2. Collection of Tannery Effluent after Treatment through CWs

Grab samples of tannery effluent were collected from each CW phytoreactor for toxicity testing, considering three different treatments, viz., CWs vegetated with *Brachiaria mutica*, *Leptochloa fusca*, and *Typha domingensis* plants only (T2), CWs augmented with consortium of salt and Cr-tolerant endophytes (*Enterobacter* sp. HU38, *Microbacterium arborescens* HU33, and *Pantoea stewartii* ASI11) while vegetated with *B. mutica*, *L. fusca*, and *T. domingensis* (T3), and CWs without vegetation or inoculation (T4). Raw tannery effluent (T5) was also collected for its toxicity comparison with treated effluents. Tap water was set as a control (T1).

### 2.3. Characterization of Treated and Untreated Tannery Effluent

In this research, standard methods to test water and wastewater as described by the American Public Health Association (APHA) were followed, for the physicochemical characterization and for the toxicity assay of tannery effluent on *C. idella* [45]. So, the physicochemical characterization of collected

effluent samples was carried out prior to the toxicity bioassay, to quantify pH, EC, $BOD_5$, COD, TDS, TSS, and Cr concentration. Results of the analysis of physico-chemical parameters of untreated and treated tannery effluent are given in Table 1.

### 2.4. Dilution of Tannery Effluent

The dilution of toxic effluents is generally required to perform the bioassays, as it is needed to reduce the effects of inherent toxic substances. Raw tannery effluent sample was diluted at different levels (25%, 50%, 75%, and 100%), using tap water. The same dilution levels were also prepared for effluent samples treated by CWs.

### 2.5. Experimental Setup

All the collected effluent samples were subjected to the toxicity test at 25%, 50%, 75%, and 100% dilution level (Table 2). So, 28 batches (in triplicates) of ten healthy fish, with average weight 260–270 mg and size 6.5 ± 0.5 cm, were introduced to each effluent sample in glass aquaria of 30 L capacity that were half filled to avoid overcrowding. As the toxicological response of the fish may vary with the body weight, and hence mortality and LC50 values may fluctuate in the same test individuals, fish of equal weight and size were employed for this assay. Four batches of 10 fish were kept in untreated wastewater at the same dilution levels, while one batch of 10 fish was kept in unamended water to be used as a control along with experimental groups. Oxygen was provided with the help of aeration pumps in each test container during the experimental period. All chemicals and reagents were analytical grade, and were provided by Sigma-Aldrich, Burlington, MA, USA and Merck, Darmstadt, Germany.

### 2.6. Toxicity Evaluation

The experiment was monitored for 96 h (exposure period), and the number of surviving fish was recorded by counting the dead fish at 24, 48, 72, and 96 h to calculate the mortality. The median lethal concentration (LC50) for 96 h, and respective upper and lower limits at 95% confidence level, were calculated as well. Acute toxicity unit (ATU) was also calculated by using the following formula: ATU = 100/LC50% (v/v).

### 2.7. Morphological and Bahavioural Response Variations

Variations in physical appearance, such as change in body colour, thickening of mucous coating on fish body, etc., were observed during the experimental period. Along with this, the behavioural changes such as swimming movements and responses of the fish to both untreated and treated effluent were observed and compared with the control set.

### 2.8. Statistical Data Analysis

The results were subjected to statistical evaluation using the Trimmed Spearman–Karber Method, Version 1.5 (US Environmental Protection Agency, Cincinnati, OH, USA). The data of each concentration was pooled to calculate the mortality, 96 h-LC50 values, and 95% confidence interval with upper and lower limits.

**Table 1.** Physico-chemical characterization of untreated and treated tannery effluent.

| Parameters | Raw Tannery Effluent | Tap Water | Effect of Different Treatments on Pollutant Removal by Constructed Wetlands (CWs) | | | | | | | CWs without Vegetation |
|---|---|---|---|---|---|---|---|---|---|---|
| | | | Tannery Effluent Treated by CWs Using *Typha domingensis* | | Tannery Effluent Treated by CWs Using *Leptochloa fusca* | | Tannery Effluent Treated by CWs Using *Brachiaria mutica* | | | |
| | | T1 | T2 | T3 | T2 | T3 | T2 | T3 | | T4 |
| Colour | Black | Colourless | Yellow | Yellow | Yellow | Yellow | Yellow | Yellow | | Brown |
| pH | 7.9 ± 0.4 | 7.9 ± 0.4 | 7.8 ± 0.3 | 7.4 ± 0.2 | 8.3 ± 0.1 | 7.8 ± 0.1 | 8.2 ± 0.1 | 7.7 ± 0.3 | | 8.5 ± 0.2 |
| EC (mS/cm) | 20.2 ± 1.3 | 1.37 ± 0.3 | 1.53 ± 0.47 | 2.07 ± 0.1 | 1.01 ± 0.01 | 0.20 ± 0.01 | 5.65 ± 0.12 | 4.64 ± 0.21 | | 8.66 ± 0.29 |
| TDS (mg/L) | 12,928 ± 1549 | 666 ± 27 | 1715 ± 39 | 1198 ± 56 | 646 ± 6.23 | 130 ± 4.5 | 3619.84 ± 45 | 2973.44 ± 25 | | 5542 ± 37 |
| TSS (mg/L) | 4800 ± 230 | - | 11 ± 0.5 | 10 ± 1.4 | 10 ± 3.5 | 9 ± 2.1 | 12 ± 0.7 | 10 ± 2.1 | | 13 ± 4.0 |
| BOD$_5$ (mg/L) | 3860 ± 612 | - | 458 ± 45 | 190 ± 18 | 392 ± 3.4 | 57 ± 1.2 | 1003.60 ± 21 | 810.6 ± 11 | | 1029 ± 25 |
| COD (mg/L) | 6066 ± 1335 | - | 760 ± 21 | 432 ± 12.5 | 728 ± 23 | 152 ± 13 | 1819.80 ± 13 | 1455.84 ± 7.5 | | 1880 ± 41 |
| Oil & grease (mg/L) | 362 ± 2.5 | - | 66 ± 5.8 | 48 ± 3 | 36 ± 2.2 | 8.3 ± 1.3 | 112.22 ± 2.2 | 97.74 ± 1.3 | | 141 ± 6.7 |
| Cr (mg/L) | 247 ± 5.8 | 0 | 2.1 ± 0.3 | 1.3 ± 0.05 | 3.3 ± 0.05 | 0.9 ± 0.01 | 1.7 ± 0.1 | 1.2 ± 0.11 | | 13 ± 1.43 |

T2 = Plants; T3 = Plants + bacteria; T4 = Without plants. Each value is mean ± standard deviation and number of replicates for each treatment (*n*) = 3. EC = Electrical conductivity, TDS = Total dissolved solids, TSS = Total suspended solids, BOD$_5$ = Five-day biochemical oxygen demand, COD = Chemical oxygen demand, Cr = Chromium, pH = Pondus Hydrogenii.

**Table 2.** Mortality of *Ctenopharyngodon idella* exposed to different concentrations of both untreated and treated tannery effluent.

| Effluent Concentration % | | Mortality of *C. idella* in Untreated and Treated Tannery Effluent after 96 h Exposure Period (%) | | | | | | | | |
|---|---|---|---|---|---|---|---|---|---|---|
| | | Treated Effluent by CWs Using *T. domingensis* | | | Treated Effluent by CWs Using *L. fusca* | | | Treated Effluent by CWs Using *B. mutica* | | |
| | | T2 | T3 | T4 | T2 | T3 | T4 | T2 | T3 | T4 |
| 0 (Control) | | 0 | 0 | 0 | 0 | 0 | 0 | 0 | 0 | 0 |
| 25 | | 20 | 0 | 40 | 0 | 0 | 40 | 20 | 10 | 40 |
| 50 | | 30 | 0 | 80 | 10 | 0 | 80 | 50 | 20 | 80 |
| 75 | | 50 | 0 | 100 | 20 | 10 | 100 | 80 | 40 | 100 |
| 100 | | 60 | 10 | 100 | 20 | 10 | 100 | 90 | 60 | 100 |
| 96 h-LC50 | | 76.61% | - | 29.73% | - | - | 29.73% | 46.53% | 86.60% | 29.73% |
| 95% Confidence limit | Lower limit | 39.18 | - | 19.59 | - | - | 19.59 | 33.27 | 63.13 | 19.59 |
| | Upper limit | 149.79 | - | 45.13 | - | - | 45.13 | 65.07 | 118.68 | 45.13 |
| 96 h-LC50 as ATU | | 1.30 | - | 3.36 | - | - | 3.36 | 2.14 | 1.15 | 3.36 |

T1 = Control; T2 = Plants; T3 = Plants + bacteria; T4 = Without plants; T5 = Mortality in raw tannery effluent at any dilution was 100%. LC = Lethal concentration inducing 50% mortality of *C. idella*, ATU = Acute toxicity unit.

## 3. Results

Fish were exposed to untreated and treated tannery effluent using different concentrations for a short-term (96 h) exposure period. Characterization of raw effluent in terms of physico-chemical parameters revealed relatively much higher concentrations of pollutants than treated effluent, as shown in Table 1. In addition, general observations on fish morphology, behaviour, mortality, and toxicity were also made.

### 3.1. Mortality and Toxicity Evaluation

The experimental species, *C. idella,* showed differential toxicity level with varying exposure time. The observed percentage of mortality was recorded for 24, 48, 72, and 96 h, as shown in Figures 1–3.

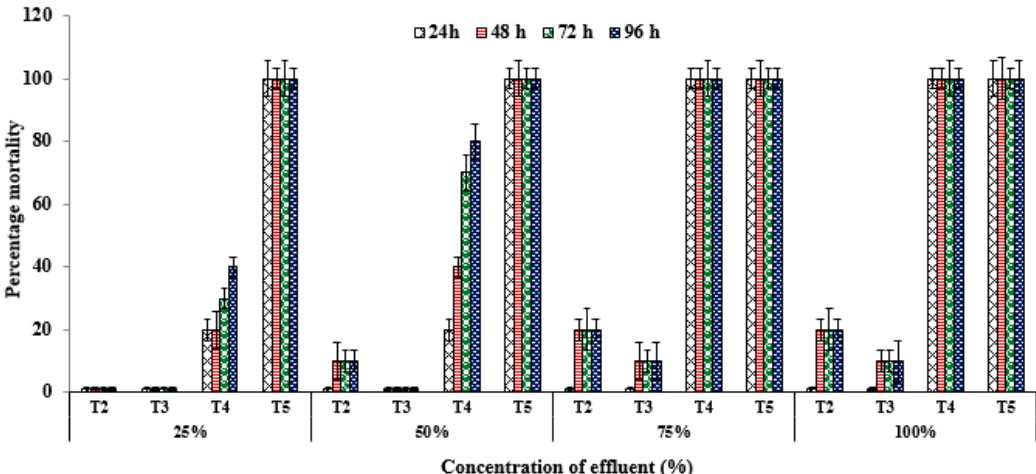

**Figure 1.** Mortality in fish on exposure to different concentrations of untreated and treated effluent by CWs using *Typha domingensis.* Tannery effluent treated in CWs with only *T. domingensis* (T2), effluent treated in CWs with *T. domingensis* and bacteria (T3), effluent treated in CWs without vegetation (T4), and raw tannery effluent (T5). Each value is mean ± standard error and number of replicates for each treatment (*n*) = 3.

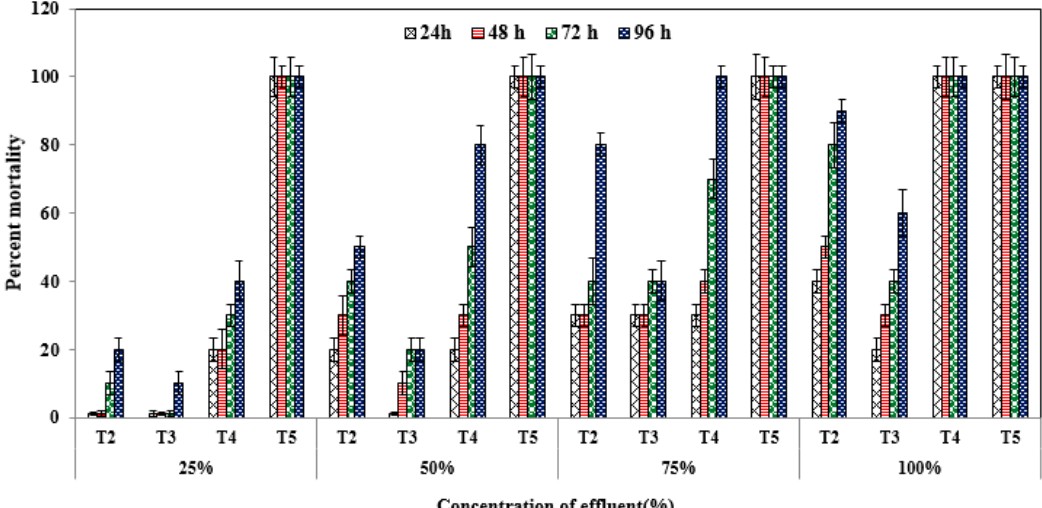

**Figure 2.** Mortality in fish on exposure to different concentrations of untreated and treated effluent by CWs using *Leptochloa fusca.* Tannery effluent treated in CWs with *L. fusca* only (T2), effluent treated in CWs with *L. fusca* and bacteria (T3), effluent treated in CWs without vegetation (T4), and raw tannery effluent (T5). Each value is mean ± standard error and number of replicates for each treatment (*n*) = 3.

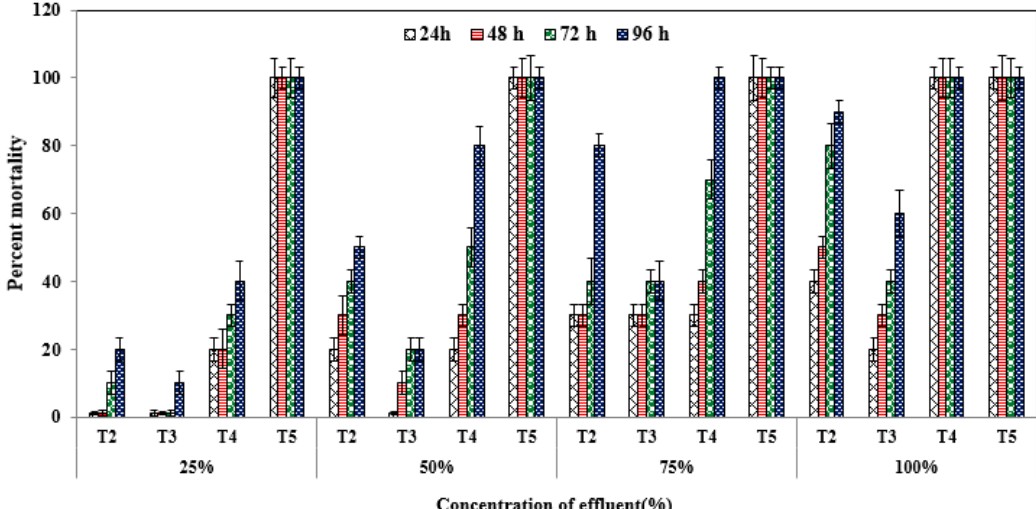

**Figure 3.** Mortality in fish on exposure to different concentrations of untreated and treated effluent by CWs using *Brachiaria mutica.* Tannery effluent treated in CWs with *B. mutica* only (T2), effluent treated in CWs with *B. mutica* and bacteria (T3), effluent treated in CWs without vegetation (T4), and raw tannery effluent (T5). Each value is mean ± standard error and number of replicates for each treatment (*n*) = 3. Stocked in T3 (effluent treatment by combined use of *L. fusca* and bacteria) at 50% effluent concentration, while the maximum percentage mortality observed was only 10% in higher effluent concentrations (100%). Observations for T4 treatment were the same as described above.

### 3.2. Toxicity of Raw Tannery Effluent

Raw effluent caused high acute toxicity and 100% mortality in exposed fish, even at dilutions up to 25% (Table 2). The fish died immediately after release to raw effluent (T5). In contrast, no mortality of fish was observed in tap water that was set as a positive control (T1).

### 3.3. Toxicity of Effluent Treated by CWs Using T. domingensis

The toxicity study of effluent treated by CWs vegetated with *T. domingensis* only (T2) revealed that the mortality of fish was 30% at the 24 h exposure period and increased to 60% at reaching the 96 h exposure period, while its dilution up to four times decreased fish mortality up to 20%, as shown in Figure 1. Moreover, combined use of plants and bacteria (T3) decreased the mortality of fish up to 10% at the 96 h exposure period, and its two times dilution further decreased this rate to zero. But the effluent treated in CWs in the absence of vegetation (T4) showed high mortality (100%) that decreased to 40% at 1:3 dilutions, while its LC50 value was 30% (v/v), with 19.59 highest and 45.13 lowest limits at 95% confidence level, and ATU was 3.36, revealing high acute toxicity (1–10).

### 3.4. Toxicity of Effluent Treated by CWs Using L. fusca

The toxicity assay conducted on effluent treated in CWs by using *L. fusca* plant and effects of T2, T3, and T4 treatments on percentage mortality of the fish for the 96 h exposure time is shown in Figure 2. As indicated by this figure, T2 (CWs with *L. fusca*) treatment showed 20% mortality at 100% effluent concentration, whereas reduced to 10% at 50% effluent concentration, and no mortality was observed at 25% effluent concentration. Among all the treatments, the lowest mortality was observed on the fish.

### 3.5. Toxicity of Effluent Treated by CWs Using B. mutica

Figure 3 shows the percentage mortality of the fish exposed to effluent treated by CWs vegetated with *B. mutica* (T2), revealing that 40% of fish died after the 24 h exposure period, and at reaching 96 h of exposure, 50% more fish died in this effluent. However, its dilution up to 25% decreased the death

rate, and 80% of fish survived until the 96 h exposure time. While effluent treated by augmenting CWs with bacteria along with *B. mutica* (T3) reduced the mortality percentage from 90% to 60% at 96 h exposure duration and 100% effluent concentration, while its dilution up to 25% decreased the death rate from 60% to 10%. The 96 h-LC50 was also extrapolated to be 46.53% (v/v), with the highest and lowest levels being 33.27 and 65.07 respectively, and 2.14 as ATU for T2 (Table 2). While for T3, 96 h-LC50 was 86.60% (v/v), with 63.13 upper and 118.68 lower limits, and 1.15 as ATU (Table 2). The confidence limit was 95%. Acute toxicity results show a comparatively low mortality level at lower effluent concentrations, while it was higher in concentrated effluent.

### 3.6. Relative Performance of Different Plant Species in Toxicity Reduction of Tannery Effluent by CWs

Comparative analysis of different plant species to reduce toxicity of tannery effluent by CW technology is presented in Figure 4. The maximum toxicity of effluent was reduced by *L. fusca*, followed by *T. domingensis* and *B. mutica*. Tannery effluent treatment in CWs with vegetation and bacterial augmentation resulted in more reduced toxicity than vegetation alone in CWs.

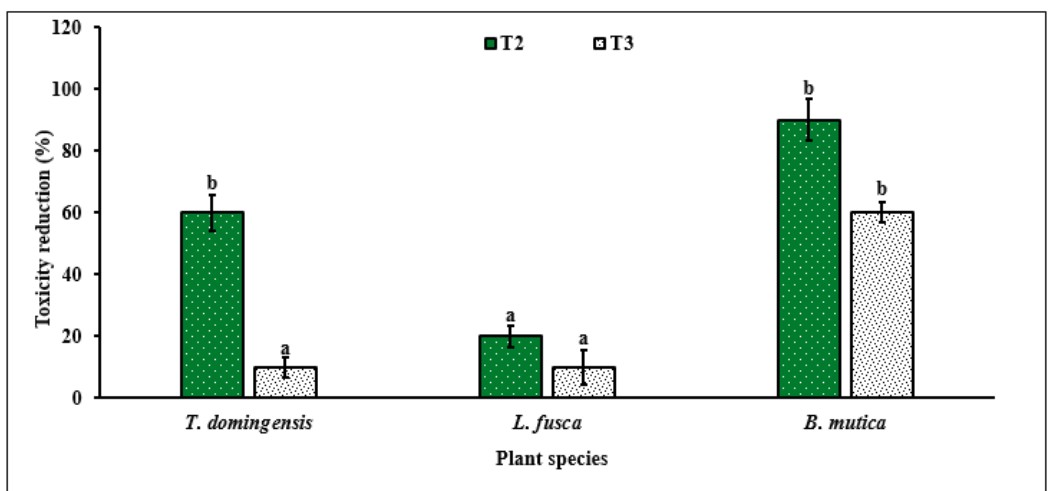

**Figure 4.** Relative performance of different plant species in toxicity reduction of tannery effluent in constructed wetland systems. Tannery effluent treated in CWs with plants only (T2), and with plants and bacteria (T3). Labels (a) and (b) indicate statistically significant differences ($p < 0.05$) among plant species for toxicity reduction at a 5% level of significance. Each value is mean ± standard error and number of replicates for each treatment ($n$) = 3. Different letters on each error bar show significant differences among different treatments.

### 3.7. Morphological Changes

Morphological change in fish colour was observed on exposure to untreated tannery effluent (T5) and effluent treated by CWs without vegetation (T4), i.e., body of fish became dark, as shown in Figure 5. In addition to this, a thick coat of mucus, spreading all over the body of the fish, was also observed in T4 and T5 treatments, making fish slimier. In effluent treated by CWs with plants only (T2), and with combined use of plants and bacteria (T3), as well as in control (T1), there was apparently no change in body colour, and fish had a normal mucus coating on their body.

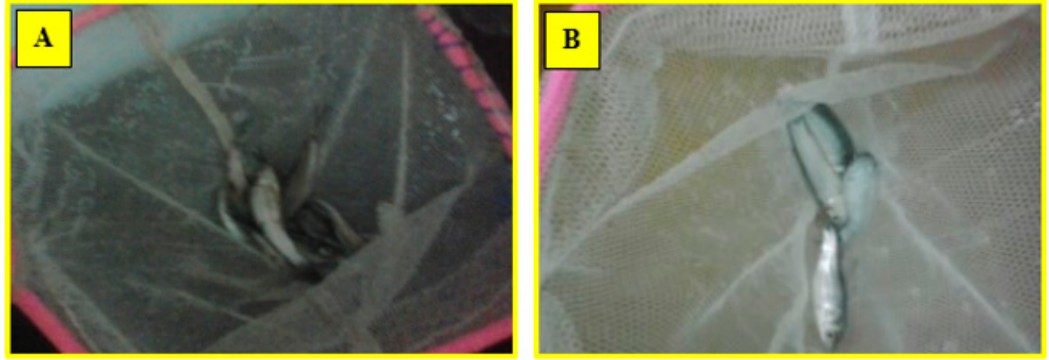

**Figure 5.** Morphological change in *C. idella* on exposure to tannery effluent before (**A**) and after treatment (**B**) by CW technology.

*3.8. Variations in Behavioural Response*

On the basis of visual observations, data regarding behavioural response of *C. idella* were recorded. As the experiment progressed, fish exhibited irregular zig-zag movements, frequent surfacing, coughing, and opercular movements on exposure to untreated tannery effluent (T5) and effluent treated without vegetation in CWs (T4), in contrast to normal movements and swimming behaviour without any loss of equilibrium in the rest of treated effluent samples and in tap water set as a control (Figure 6).

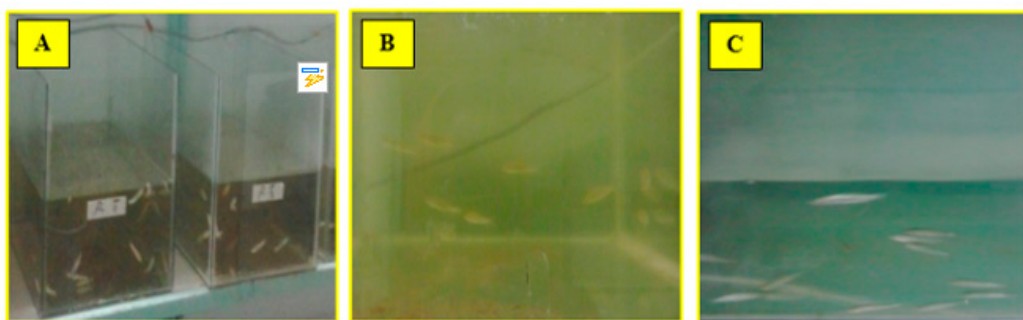

**Figure 6.** Variation in behavioural response of *C. idella* on exposure to untreated tannery effluent at 50% dilution showing irregular swimming and zig-zag movements (**A**), and normal movements and swimming behaviour in treated tannery effluent at the same dilution level (**B**), and in tap water set as control (**C**).

*3.9. Physiological Variations*

Untreated tannery effluent was also loaded with oil and grease to the tune of 362 mg/L, in addition to other pollutants. This created respiratory problems in fish by lowering the dissolved oxygen in test media, despite the aeration system. It was observed that fish regularly came to the surface of water for atmospheric oxygen. Such an anomaly was not observed in T2, T3, and the control set, maintained under identical experimental conditions.

## 4. Discussion

The present investigation evaluated the toxicity of both treated and untreated tannery effluent by measuring mortality, LC50, and acute toxicity in fish for 96 h exposure duration. Along with this, morphological changes and behavioural response of the fish were also observed. In the current study, the mortality of *C. idella* was directly proportional to the exposure period, concentration of pollutants in effluent, and acute toxicity unit, as already observed in previous works [46–48]. At 96 h-LC50 concentration, the ability of natural population of aquatic bodies would be relatively impaired, and an increase in concentration also increased the mortality [49,50].

As indicated in Table 2, from 96 h-LC50 values (29.73–86.60%) at different effluent concentrations, it is clear that the raw tannery effluent exerts high toxicity, while effluent treatment using CWs technology reduced the overall pollution level to a great extent. However, LC50 values still provide information for gross comparison of toxicity of the pollutants to the fishes. The present study revealed that raw and treated effluent by using CWs without vegetation caused more or less similar toxicity, whereas combined use of plants and bacteria in CWs caused a significant reduction in toxicity level of effluent, and among all the tested plants, *L. fusca* was more efficient. The same toxicity results were reported for raw pharmaceutical effluent acting on *Lebistes reticulates* [51]. The variation in 96 h-LC50 of Cr for different species was previously reported, being 34 mg/L for *Cirrhinus mrigala* [52], 39.4 mg/L for *Labeo rohita* [53], 85.7 mg/L for *Carassius auratus* [54], 87.96 mg/L for *Cyprinus carpio* [55], and 53.08 mg/L for *C. idellus* [50].

In the present study, acute toxicity values were high in concentrated effluents as compared to diluted ones, due to varying amounts of Cr. These toxicity values are organism- and age-specific [12,56]. Here, exposure of *C. idella* to untreated effluent caused death within 24 h, while in treated effluent, most fish survived until 96 h, varying with different treatments and concentration levels of effluent. Similar findings were observed by Arora et al. [31], who assessed toxicity of treated urban wastewater and its reutilization for aquaculture on sucker fishes for 96 h and compared it with untreated wastewater. Fish died within 48 h in untreated wastewater while they survived until 96 h in the treated one. In another investigation, the acute toxicity test revealed about an 80% and 100% survival rate of *Poecilia* sp. in 100% untreated effluent and ceramic membrane-treated effluent respectively, after a 72 h exposure period [39].

Morphological changes in fish occurred in those that were exposed to untreated tannery effluent (T5) and effluent treatment by CWs without vegetation (T4). These changes were darkening of body colour and a thick mucus coating spreading all over the body, due to the presence of toxins in untreated effluent. The same findings were reported by Carpenter [57,58] and Nisha et al. [59]. Thus, these facts are directly reflecting the toxicity level of effluent, because no such morphological change in fish was observed in T2, T3, and the control set (T1). According to Handa and Jindal [50], Cr-contaminated water has the potential to induce morphological and behavioural changes in *C. idellus.* Chromium toxicity also affects the normal survival of fish and produces allergic reactions on skin [60]. It also impairs functioning of the nervous system [61], which causes darkening of body colour due to the reversible response of melanophores [62].

Behavioural responses, the most perceptive indicator of toxicity, are crucial for normal life functions, and ultimately fish survival [63]. The current study revealed that lethal concentrations of tannery effluent caused abnormal changes in the fish, and intensity increased with time. Similar effects were observed on the behaviour of *Mystus vittatus* and *L. reticulates* on exposure to paper mill and drug industry effluent, respectively [64,65]. Related behavioural changes, such as jerky movements, opening of mouth, erratic swimming, and secretion of mucus, etc., were also observed by Timchalk et al. [66], Bantu and Vakita [67], and Nisha et al. [59].

Exposure of fish to raw tannery effluent proved to be highly unfit for their survival, as fish became inactive immediately after the addition of effluent. Frequent surfacing of fish to gulp the oxygen at initial stages, followed by slow erratic swimming and zig-zag irregular movements, as well as loss of balance, and finally settling at the bottom of the aquarium after a few hours, was an indication of toxicity defects caused on fish due to raw tannery effluent. Similar distressed behavioural observations were also reported earlier [2,59,68–70]. This reduced normal activity could be a consequence of depletion of energy in the body of fish due to impairment of carbohydrate metabolism, wherein organisms that could not tolerate the contaminants enter into a state of coma and subsequent death [47,71]. Cui et al. [72] also noticed a decrease in fish movement leading to cessation of swimming [73] due to the altered carbohydrate metabolism and neurotoxicity induced by Cr metal [74]. According to Roopadevi and Somashekar [47], fish lose their ability to maintain homeostasis on exposure to higher

effluent concentrations for a longer period, and this eventually causes mortality, or some physiological stress may be the reason, which is confirmed by the present results.

Previous investigations also reported similar observations when fish were exposed to toxic industrial effluents, being irregular hyper and hypoactive movements, loss of balance, and increase in surfacing activity that ultimately decrease their tolerance to high pollution load [75–77]. The present findings are also in concordance with earlier studies of fish behavioural response to industrial effluents, revealing normal behaviour and swimming of fish in control, in contrast to the abnormal swimming, loss of equilibrium, coughing, and opercular movements in toxic effluents [2,59,70,78]. According to Sinha and Kumar [79], equilibrium loss and abnormal swimming are caused by damage to the nervous system that regulates all the vital functions.

High organic pollution load of tannery effluent increases its $BOD_5$ and subsequently reduces its dissolved oxygen level. This oxygen deficiency caused respiratory problems in fish, along with this reduced tolerance capacity of fish towards toxins that caused death. Oyedapo and Akinduyite [80] similarly reported that toxic effluents disrupt the ability of fish to maintain homeostasis and cause physical damages. Another reason for a reduction in oxygen consumption may be malfunctioning of gills due to direct contact with pollutants present in raw tannery effluent. Previous studies support this evidence, indicating that high levels of toxic substances in the test medium cause coagulation of gill mucous in fish that alters their respiratory metabolism [47,49,77,81]. Moreover, accumulation of mucus on the gills of fish and asphyxiation decrease oxygen consumption. A similar kind of observation was also reported by Thorat and Wagh [82], who showed that tanning industry effluent has a strong effect on the oxygen consumption rate of fish, *Channa gachua*, probably due to Cr toxicity.

Toxicity assays play a viable role in monitoring the pollutants' level in industrial effluents and can be used as a promising technology to regulate quality of industrial discharge to receiving reservoirs. As discussed earlier, fish show changed behaviour on exposure to a toxic environment, an indicator of the surrounding pollution level. Such types of variations in behavioural response of fish are an easy way to directly assess the quality of industrial discharges.

Biotoxicity tests are an economical and technical method for direct measurement of toxicity of industrial effluents. In addition to this, the present study showed that toxicity bioassays must be performed to assess the toxicity of treated effluent along with physico-chemical analyses, such as pH, EC, COD, $BOD_5$, and TSS, to guarantee the aquatic organisms' safety and minimise ecotoxicological issues.

## 5. Conclusions

*Ctenopharyngodon idella* were seriously affected by the tested raw tannery effluent, even at very low concentrations, despite making dilutions due to high toxicity of pollutants, in contrast to no adverse effects in treated effluent by CW technology and the control set. Effluent treatment by CWs vegetated with different plants (*B. mutica*, *L. fusca*, and *T. domingensis*) reduced the toxicity causing abnormal changes in morphology and physiology of fish that ultimately lead to mortality, and inoculation of endophytic bacteria (*Enterobacter* sp. HU38, *Microbacterium arborescens* HU33, and *Pantoea stewartii* ASI11) further caused a reduction in its toxicity level up to 90%. In contrast to these treatments, CWs without vegetation could not reduce the toxicity of this effluent. Hence, fish survived for longer periods in effluent treated by CWs vegetated with *L. fusca* plants and augmented with endophytes, wherein pollutants were detoxified by the synergistic effect of plant–microbe interactions. In addition to this, treatment performance of *L. fusca*-planted CWs was more effective than *B. mutica*- and *T. domingensis*-planted CWs. Therefore, this study emphasizes the use of CWs technology to treat tannery effluent, preferably planting *L. fusca* species, along with bacterial augmentation that ultimately results in the reduction of effluent toxicity and aquatic pollution and poisoning. The current bioassay results illustrated that disposal of raw tannery effluent is unsafe to fish, causing mortality, alterations in morphology, physiology, and behaviour, due to its inherent toxicity. Therefore, treatment of tannery effluent by nature-friendly CWs technology is highly recommended prior to its discharge to water

resources (by mixing 50% fresh water) to protect the environment. Moreover, toxicity assays on fish can be practiced economically to ensure safe disposal of industrial discharge.

**Author Contributions:** Conceptualization, S.A. (Sobia Ashraf), M.A., M.N.; methodology, S.A. (Sobia Ashraf), S.A. (Sana Ashraf), and K.R.; software, S.A. (Sobia Ashraf), and M.A.; validation, M.N., Z.A.Z., A.N.-D., S.R.A., and M.A.; formal analysis, S.A. (Sobia Ashraf), K.R., and M.A.; investigation, S.A. (Sobia Ashraf), M.N., M.A.; resources, M.A., Z.A.Z., A.N.-D., and M.N.; data curation, S.A. (Sobia Ashraf), K.R., S.A. (Sana Ashraf); writing—original draft preparation, S.A. (Sobia Ashraf), and M.A.; writing—review and editing, M.A., Z.A.Z., M.N., S.R.A., and A.N.-D. All authors have read and agreed to the published version of the manuscript.

**Funding:** This research was conducted under the grant number, No. 20-3854/R&D/HEC/14., of the Higher Education Commission (HEC), Pakistan.

**Acknowledgments:** The authors are thankful to the Institute of Soil and Environmental Sciences, University of Agriculture, Faisalabad 38000, Pakistan, and the National Institute for Biotechnology and Genetic Engineering (NIBGE), Faisalabad, Pakistan, for providing research facilities.

**Conflicts of Interest:** The authors declare no conflict of interest.

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
