# Peer review of "Evaluation of Toxicity on Ctenopharyngodon idella Due to Tannery Effluent Remediated by Constructed Wetland Technology"

_processes, doi:10.3390/pr8050612_

Round 1

Reviewer 1 Report

This study used bioassays to evaluate the toxicity of tannery effluent on Ctenopharyngodon idella after constructed wetland treatment. The topic of this research is interesting. Some questions and suggestions are offered below with the intent to assist the author in improving the manuscript.

  1. It is recommended to reduce the narrative of the research background in the abstract and increase the description of the research results.
  2. Please use (A), (B) to mark the two small images of Figure 5.
  3. Figure 5 can't clearly see the C. idella body condition as described in the manuscript, for example, the color of the fish body changes simultaneously with the lightness and darkness of the overall photo. Therefore, the author is requested to provide a clear picture of the fish body with the same brightness.
  4. Can the author provide any image data in Sections 3.8 and 3.9?
  5. This study used tap water as the control group (T1). Although there is no fish died in the control group, tap water should also be analyzed for Physico-chemical characteristics. Therefore, the author is requested to include the analysis data of Tap water in Table 1.
  6. English expression needs to be improved and compact description of academic in whole manuscript is warranted.

Author Response

Manuscript ID: processes-793929

Response to comments of Reviewer 1:

This study used bioassays to evaluate the toxicity of tannery effluent on Ctenopharyngodon idella after constructed wetland treatment. The topic of this research is interesting. Some questions and suggestions are offered below with the intent to assist the author in improving the manuscript.

 Comment 1: It is recommended to reduce the narrative of the research background in the abstract and increase the description of the research results.

Response: Many thanks for your kind suggestion and abstract of the manuscript has been modified accordingly. (line 19-23; 35-38)

Comment 2: Please use (A), (B) to mark the two small images of Figure 5.

Response: Thank you for your suggestion. The said correction has been made in the revised version of the manuscript.

Comment 3: Figure 5 can't clearly see the C. idella body condition as described in the manuscript, for example, the color of the fish body changes simultaneously with the lightness and darkness of the overall photo. Therefore, the author is requested to provide a clear picture of the fish body with the same brightness.

Response: We agree with the reviewer and Figure 5 has been replaced in the revised manuscript with more clear vision of overall photo depicting the color of fish body.

Comment 4: Can the author provide any image data in Sections 3.8 and 3.9?

Response: Figure 6 has been added in section 3.8 in the revised manuscript showing variation in behavioural pattern of fish on exposure to tannery effluent.

Comment 5: This study used tap water as the control group (T1). Although there is no fish died in the control group, tap water should also be analyzed for Physico-chemical characteristics. Therefore, the author is requested to include the analysis data of Tap water in Table 1.

Response: We agree with the reviewer that tap water used for the experiment should be analyzed for physico-chemical characteristics parameters. Therefore, the analysis data of tap water has been included in Table 1 in the revised manuscript.

Comment 6: English expression needs to be improved and compact description of academic in whole manuscript is warranted.

Response: Thank you for your comment. We have tried our best to improve the overall manuscript.

Reviewer 2 Report

Overall In my opinion it is a good job, with a good experimental design and written well enough. However, I must underline some shortcomings and modifications that in my opinion would make the work more complete and more fluid for the reader.

First of all, the introduction is well written and fairly focused on the problem of publication, although from line 77 to line 102 the list of previous works examined is not very fluid and looks more like a list, in my opinion it would have made it more fluid. Furthermore, one of the focal points of the work are the constructed wetlands (CWs) of which very little is spoken. I suggest adding more information about the structure, plant and bacterial species that are currently used in these phyto-reactors.

Again with regard to the treatment CWs which in materials and methods are described from line 118 to 122, in my opinion it would be better to spend a few more words, for example on the bacterial consortium that is associated with plants in the T3 treatment.

Moving on to results only a form advice, starts from line 169 up to 225 sub-paragraphs are not necessary, in my opinion is better to organise it as a single draft to make the reading more fluent.

The same from line 231 to 249.

Discussions are focused and well written.

In the conclusions of the work I would propose to emphasize mainly on what are the scientific innovations brought about this publication that do not turn out well.

Author Response

Manuscript ID: processes-793929

Response to comments of Reviewer 2:

Overall, in my opinion it is a good job, with a good experimental design and written well enough. However, I must underline some shortcomings and modifications that in my opinion would make the work more complete and more fluid for the reader.

Comment 1: First of all, the introduction is well written and fairly focused on the problem of publication, although from line 77 to line 102 the list of previous works examined is not very fluid and looks more like a list, in my opinion it would have made it more fluid. Furthermore, one of the focal points of the work are the constructed wetlands (CWs) of which very little is spoken. I suggest adding more information about the structure, plant and bacterial species that are currently used in these phyto-reactors.

Response: Thanks for your remarks. The said suggestions have been incorporated in the revised version of the manuscript.

Comment 2: Again with regard to the treatment CWs which in materials and methods are described from line 118 to 122, in my opinion it would be better to spend a few more words, for example on the bacterial consortium that is associated with plants in the T3 treatment.

Response: Thank you for your comment. The detail about the bacterial consortium used in the T3 treatment by CWs has been added in the revised manuscript. (Line 123-124)

Comment 3: Moving on to results only a form advice, starts from line 169 up to 225 sub-paragraphs are not necessary, in my opinion is better to organise it as a single draft to make the reading more fluent.

Response: We are thankful to you for suggesting a valuable advice. Here, we divided different parts of the results into subparagraphs just to make it more comprehensive and providing a clear description of performance of different plant species in reducing the toxicity of tannery effluent.

Comment 4: The same from line 231 to 249.

Response: Here too, from line 231 to 249 divided into subsections, for better description of variations in morphology, physiology, and behavioral pattern of fish on exposure to different levels of toxicity of tannery effluent that was treated by CWs using different plant species.

Comment 5: Discussions are focused and well written.

Response: Thank you for your appreciation.

Comment 6: In the conclusions of the work I would propose to emphasize mainly on what are the scientific innovations brought about this publication that do not turn out well.

Response: We are thankful to the you for proposing this suggestion. So, the conclusion section has been improved in the revised manuscript with emphasis on the scientific innovations of the work.

Reviewer 3 Report

The manuscript can be published.

Author Response

Manuscript ID: processes-793929

Response to comment of Reviewer 3:

Comment: The manuscript can be published.

Response: We are highly grateful for your approval of our work for publication.
